# Quantitative cytokine level of TNF-α, IFN-γ, IL-10, TGF-β and circulating Epstein-Barr virus DNA load in individuals with acute Malaria due to *P. falciparum* or *P. vivax* or double infection in a Malaria endemic region in Indonesia

**Insani Budiningsih[1], Yoes Prijatna Dachlan[2], Usman Hadi[3]\*, Jaap Michiel Middeldorp[4]\***

1 Post Graduate Doctoral Program, Faculty of Medicine, Universitas Airlangga, Surabaya, Indonesia,
2 Department of Parasitology, Faculty of Medicine, Universitas Airlangga, Surabaya, Indonesia,
3 Department of Internal Medicine, Dr. Soetomo Hospital-School of Medicine, Universitas Airlangga, Surabaya, Indonesia, 4 Department of Pathology, VU University Medical Center, Amsterdam, The Netherlands

\* usman.hadi2@fk.unair.ac.id (UH); j.middeldorp@amsterdamumc.nl (JMM)

## Abstract

*Plasmodium falciparum* Malaria and Epstein-Barr Virus (EBV) infection are risk factors in the development of Burkitt's lymphoma. In Indonesia, 100% of the population is persistently infected with EBV early in life and at risk of developing EBV-linked cancers. Currently, 10.7 million people in Indonesia are living in Malaria-endemic areas. This cross-sectional study was initiated to investigate how acute Malaria dysregulates immune control over latent EBV infection. Using blood and plasma samples of 68 patients with acute Malaria and 27 healthy controls, we measured the level of parasitemia for each plasmodium type (*P. falciparum*, *P. vivax*, and mixed) by microscopy and rapid test. The level of 4 regulatory cytokines was determined by quantitative ELISA and the level of circulating EBV genome by real-time PCR targeting the single copy EBNA-1 sequence. All Plasmodium-infected cases had high-level parasitemia (>1000 parasites/ul blood) except for one case. EBV-DNA levels were significantly more elevated in *P. falciparum* and *P. vivax* infections (P<0.05) compared to controls. EBV-DNA levels were not related to age, gender, Malaria symptoms, or plasmodium type. TNF-α and IL-10 levels were increased in Malaria cases versus controls, but IFN-γ and TGF-β levels were comparable between the groups. Only TNF-α levels in *P. falciparum* cases showed a clear correlation with elevated EBV DNA levels ($R^2 = 0.8915$). This is the first study addressing the relation between EBV (re)activation and cytokine responses during acute Malaria, revealing a clear correlation between pro-inflammatory cytokine TNF-α and EBV-DNA levels, specifically in *P. falciparum* cases, suggesting this cytokine to be key in dysregulating EBV homeostasis during acute *P. falciparum* Malaria.

**Data Availability Statement:** All relevant data are contained within the manuscript and its Supporting Information files.

**Funding:** The author(s) received no specific funding for this work.

**Competing interests:** The authors have declared that no competing interests exist.

## Introduction

Epstein-Barr Virus (EBV) is one of the most common viruses infecting mankind and persists for life in its host after the first contact. EBV primarily infects and reproduces in B-lymphocytes and epithelial cells located in the oro-nasal cavity and surrounding lymphoid tissues and circulates in a latent form in quiescent memory B-cells [1–3]. A small number of latency associated EBV gene products is essential for EBV genome maintenance and survival of infected cells, which have an inherent capacity to transform infected cells into proliferating malignant cells [1, 4]. Usually, EBV infection is under tight immune control and does not cause health problems but in certain populations and under defined conditions (cellular stress or co-factors) EBV can cause serious diseases that vary from self-limiting acute infectious mononucleosis (kissing disease) to chronic severe EBV infection, lymphoid and epithelial malignancies as well as autoimmune diseases [2, 5–7]. It is well known that disturbances in the immune response may dysregulate EBV homeostasis with chronic and potential oncogenic consequences [6–8]. EBV was first identified in cells of Burkitt Lymphoma (BL), an endemic cancer among sub-Saharan children, that is triggered by co-infection with Malaria parasites [reviewed in 8, 9]. BL is the most common cancer in children living in Malaria endemic regions in sub-Sahara Africa and Papua New Guinea [9, 10], but is also observed in children and adults with uncontrolled HIV infection [9]. During acute Malaria EBV infected memory B-cells can interact with Malaria parasites, particularly with the CIDR1α domain of *P. falciparum* Erythrocyte Membrane Protein 1 (PfEMP1), causing unregulated activation of EBV+ B-cells, lymphoproliferations and potentially leading to BL [11–13]. Children with acute Malaria have elevated EBV-DNA levels in the circulation which may return to normal upon resolving the Malaria infection [14].

In Indonesia, with around 270 million population that is 100% positive for EBV, about 10,7 million people are still living in Malaria endemic areas [15]. Children in Indonesia are exposed to EBV at early age with high dose of EBV via saliva (pre-chewed food). Later in life chronic exposure to EBV carcinogens, such as formalin, tobacco additives, herbal drugs/oils, butyrate acid (dried meat) and nitrosamine (dried salty fish) are common, which can trigger aberrant and pathogenic EBV activity and malignancy [16–18].

The immune system is a highly regulated and balanced system with neutrophils, macrophages, and NK cells acting against protozoan parasites by innate and adaptive immune responses. Innate immune cells together with dendritic cells play a vital role in the induction of T- and B-cell mediated adaptive immune responses by producing different pro-inflammatory (IL-1β, IL-6, IL-8, IL-12, IL-17, IFN-γ, and TNF-α) and anti-inflammatory (TGF-β, IL-4, IL-5, IL-10, and IL-13) cytokines that cause clinical symptoms, together resulting in parasite eradication and ultimately return to immune homeostasis [13, 19, 20]. These anti-parasite responses may affect the delicate immune balance between EBV and its host [6, 7, 11, 14, 21].

The aim of this research was to investigate how acute Malaria dysregulates EBV homeostasis and what cytokines would be involved in a Malaria-endemic population in Indonesia. Previous studies in Eastern-Africa indicated that Malaria affects EBV homeostasis in children and pregnant women [14, 21, 22] showing increased EBV-DNA loads in plasma of malaria cases compared to regional matched controls. There appeared to be a direct correlation between increases in plasma EBV viral load and progression of endemic BL, associated with increasing of *P. falciparum* antibody titers [10, 23]. To our knowledge, no prior study has described the role of inflammatory cytokines in the interaction between EBV and individual Malaria parasites *P. vivax* and *P. falciparum* during episodes of acute Malaria.

## Materials and methods

### Sample collection and Malaria parasite analysis

All necessary clearances and specific approval for this study have been obtained from the Health Research Ethics Committee of Faculty of Medicine Universitas Airlangga, Surabaya (protocol No. 278/EC/KEPK/FKUA/2020) and written informed consents were taken from all the patients at the time of sample collection. Venous blood samples were collected from Malaria cases on Sumba Island in East Nusa Tenggara,—a classified high endemic region in Indonesia [15]. Malaria cases presented with a spectrum of symptoms, including fever, headaches, nausea, paleness and conjunctival pallor. Experienced health workers visited each of the suspect malaria patients at their homes in different villages and then examined the patients on site with a Rapid Diagnostic Test (RDT; see details below). When the RDT result was positive, the patient was referred to the nearest public health center in the district (such as Public Health Centre Kori and Public Health Melolo on Sumba Island) for follow-up with clinical and microscopic blood-smear examination of the malaria status by an expert parasitologist and to obtain their questionnaire and blood plasma. The plasma of malaria patients was placed in a cool box with dry ice and shipped to the Institute of Tropical Disease, University of Airlangga, Surabaya. Upon arrival, the plasma was immediately aliquoted and frozen at –80˚C. When being used, plasma samples were thawed and stored on melting ice or in a refrigerator at +2˚-8˚C. Healthy control samples were obtained from local residents in Surabaya (East-Java) not suffering from Malaria or other acute or chronic diseases, HIV or sexually transmitted diseases. A total of 95 plasma samples from either confirmed Malaria patients with positive parasites of *P. falciparum* (n = 26), *P. vivax* (n = 28), and mixed (*P. falciparum* and *P. vivax*) (n = 14), or healthy controls (n = 27) were used in this study. Of the 68 Malaria cases, 42 cases were male and 26 were female, whereas of the controls 12 were male and 15 were female. The mean age of malaria cases was 20.2 years (range 4–78) and for the healthy controls this was 29.5 (range 20–50). All samples were aliquoted and stored at -20˚C until use. The diagnosis of Malaria was confirmed by demonstrating the presence of plasmodial parasite infection in fresh blood (finger prick) using a test for Malaria antigen detection, i.c. "Rapid Diagnostic Test" (RDT) [CareStart[TM] Malaria Pf/PAN (HRP2/pLDH) Ag Combo RDT, lot.nr. RMRM-01071, ACCESSBIO, Somerset, NJ, USA]. All cases with a positive RDT were followed-up and confirmed in the regional health center (s) on Sumba Island by further blood examination using thick smear microscopy by expert parasitologists to confirm the parasite species and to quantify the proportion of infected red blood cells in relation to a predetermined number of white blood cells (WBC), according to WHO-2010 guidelines [24]. Briefly, a small blood drop was used for preparing thick smears on a glass slide for laboratory examination using oil-immersion microscopy. Giemsa-stained thick blood smears were visualized under the light microscope for the identification of various species of Malaria parasites. Parasites were counted for every 500 WBCs in each blood smear which is inferred from the number of WBC per µL of blood automatically calculated using blood cell counters or assumed at a fixed value of 8,000 cells/µL, according to the WHO-2010 guidelines [24]. The final number of parasites per µL of blood was calculated as the formula: [(counted parasites/500WBC) x counted or assumed WBC/µL] [25]. Parasitemia level were categorized as 4 groups, such as Group 1 with + = 1–10 parasites per 100 oil-immersion thick film fields, Group 2 with ++ = 11–100 parasites per 100 oil-immersion thick film fields, Group 3 with +++ = 1–10 parasites per single oil-immersion thick film field, Group 4 with ++++ = more than 10 parasites per single thick film field.

## Plasma EBV-DNA quantification

Plasma samples were processed for molecular analyses by Real Time PCR (RT-PCR) at the Institute of Tropical Disease, Universitas Airlangga, Surabaya. Total DNA was extracted using the QIAamp DNA Mini Kit (cat. nos. 51304, Qiagen, Germany) and analysed by the Epstein-Barr Virus (EBV) RT-PCR kit using external standard ISEX calibration and UNG-dUTP contamination control (Geneproof, EBV/ISEX/100, Czech Republic). This PCR is targeting the single copy DNA sequence encoding EBNA1 ensuring exact EBV genome quantification. The ISEX-sample consisted of 50 μl sample DNA eluate spiked with 5 μl Internal Standard (IS). For each PCR run, 10 μl of ISEX-sample or 10 μl of Calibrator/Positive Control or water were added into individual PCR tubes containing 30 μl of dUTP nucleotide MasterMix with uracil-N-glycosylase (UNG) for elimination of PCR product carry-over. The final reaction mix volume was 40 μl. Amplification was done in a RotorGene thermal cycler (Qiagen). Cycling conditions for the first step included the one hold step at 37˚C for 2 min followed by one hold step at 95˚C for 10 min for UNG inactivation. The cycling conditions for second step included 45 cycles of an initial denaturation at 95˚C for 5 second followed by 45 cycles of 40 second annealing at 60˚C and 20 second at 72˚C for final extension. Data were analysed and quality controlled by Rotor-Gene ScreenClust HRM Software (Qiagen). The interpretation of positive EBV viral load could be seen in the FAM and HEX channels as valid results. The formula of Sample Concentration (copies/μl) x Elution Volume (μl) / Isolation Volume (ml) was used to calculate the virus concentration in copies/ml.

## Plasma cytokine quantification

In each plasma sample, four different cytokines were quantified by capture-ELISA technique using commercial assays: Human TNF-α by RAB0476-1KT, Lot. No. 1125F0193 (Millipore, Sigma-Aldrich, Missouri-USA), human TGF-β 1 by RAB0460-1KT, Lot. No. 0127F0188, (Millipore, Sigma-Aldrich, Missouri-USA), human IL-10 by E0102Hu (Bioassay Technology Laboratory (BT-Lab), Shanghai-China) and human IFN-γ by E0105Hu (Bioassay Technology Laboratory, Shanghai-China), according to the manufacturer's protocols.

## Statistical analysis

Data collection and statistical analysis was done using Excel and Graphpad Prism version 8.0 software. The comparison between EBV genome and Malaria parasites levels in a population was done by unpaired student t-test, the correlation between gender or age and EBV-DNA load or cytokine levels was analysed for all Malaria subgroups and controls by one-way ANOVA and linear regression, respectively, and the correlation between EBV-DNA loads and individual cytokine levels was analysed by the R-square (Pearson) method.

## Results

### Parasitemia level and symptoms

All Malaria plasma samples (n = 68) were obtained from symptomatic patients living on Sumba Island, a malaria-endemic region in Indonesia [15] and were categorized as having group-4+ high-parasitemia according to WHO-2010 criteria, with more than 10 parasites per single thick film microscopic field (>1000 parasites/ul), except for one case (Fig 1). By expert microscopic analysis of thick blood smears, samples were characterized on site for plasmodium subtype, yielding specimens with *P. falciparum* (n = 26), *P. vivax* (n = 28), and mixed (*P. falciparum and P. vivax;* n = 14) infections. Highest parasite levels were found in 7/26 *P. falciparum* cases (>10,000 parasites/μl). Symptoms of Malaria cases are detailed in S1 Fig and overall show no significant differences between the different parasite groups (P = 0.0838), except that incidence of conjunctiva pallor was

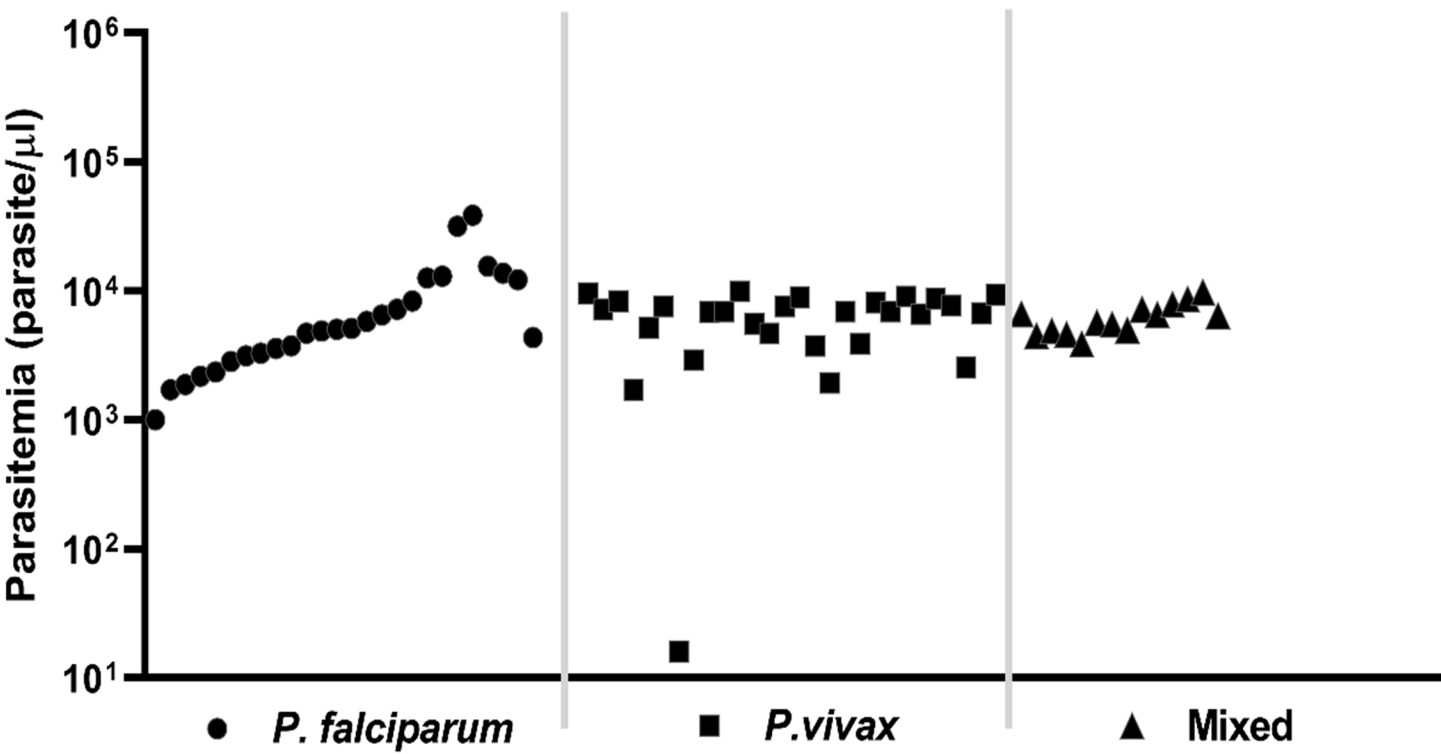

**Fig 1. Parasitemia levels of Malaria cases infected with *P. falciparum* (n = 26), *P. vivax* (n = 28) and Mixed parasites (n = 14).**

less in *P. falciparum* (50%) compared to *P. vivax* or Mixed cases (both 100%) and severe headaches were more reported in *P. falciparum* compared to *P. vivax* cases (100% vs 46%). Male/female ratios were comparable between Malaria cases and controls (M/F ratio: 42/26 and 12/15, respectively; P >0.3). Healthy controls (n = 27) were collected in the city of Surabaya, East-Java and did not have a recent inflammatory disease history and showed no signs of illness at the time of sampling.

## EBV genome by quantitative PCR (Q-PCR)

A total of 68 Malaria cases and 27 healthy controls could be analyzed for EBV-DNA genome level in blood plasma using a commercial EBNA1-targeted quantitative PCR (Fig 2). Compared to the healthy controls (mean EBV DNA level = 7,2 x$10^3$ copies/ml; SD = 2,2 x$10^4$), cases of *P. falciparum* (4.4 x$10^5$ copies/ml; SD = 9,9 x$10^5$) and *P. vivax* infection (4,6 x$10^5$ copies/ml; SD = 9,1 x$10^5$) had significantly higher mean EBV DNA levels (P = 0.0308 and 0.0142, respectively; Unpaired t-test). In Malaria patients with Mixed infection the mean EBV-DNA level was elevated compared to controls but did not reach significance (mean level = 1,2 x$10^4$ copies/ml; SD = 1,4 x$10^4$; P = 0,4666). For all patients analysed per defined malaria subgroup there was no correlation between gender or age and the level of EBV-DNA in plasma as defined by one-sided ANOVA and linear regression analysis, respectively (S2 and S3 Figs). In the healthy controls, nine individuals (33,3%) had substantially elevated EBV-DNA levels without apparent symptoms (Fig 2), but there was no relation between gender or age with EBV-DNA levels.

## Parasitemia versus EBV genome levels

For comparison of whole blood parasitemia levels with EBV-DNA load in plasma, we grouped the Malaria cases into 3 subgroups of low (<1000/ul), intermediate (1000–10.000/ul) and high

(>10.000/ul) level. No significant relation was found between parasitemia level and EBV-DNA load for the 3 subgroups of Malaria cases (P = 0,6826; P = 0,9570; P = 0,3799, respectively) (Fig 3A–3C).

## Cytokine levels in plasma by quantitative ELISA

**TNF-α.** The plasma TNF-α cytokine level was significantly increased in most Malaria case in all 3 Malaria groups, with P < 0,05 overall compared to the healthy controls (P = 0,0082, P = <0,0001, P = <0,0001, respectively; Fig 4A). The increase of TNF-α levels was seen in both male and female Malaria cases, but there was no relation with age. Some healthy controls (N = 4) showed increased TNF-α levels, but this was not related to age, gender or any specific symptoms nor to elevated EBV-DNA levels (S4 Fig).

**IL-10.** A low and variable but significantly increased anti-inflammatory IL-10 cytokine response was found in *P. falciparum* and mixed infection cases with P < 0,05 compared to the healthy controls, whereas *P. vivax* cases showed elevated levels as well (P = 0.0764). In the healthy controls IL-10 levels were nearly undetectable (Fig 4B).

**IFN-γ.** Pro-inflammatory IFN-γ cytokine levels were considerably elevated in all groups, but otherwise not significantly different between Malaria cases and healthy controls (P = 0,2398; P = 0,0842; P = 0,2832, respectively) (Fig 4C). IFN-γ levels were not related to age or gender in any of the groups.

**TGF-β.** Increased levels of anti-inflammatory cytokine TGF-β were detectable in all 3 Malaria groups compared to healthy controls, reaching significance in *P. falciparum* and mixed cases (p = 0.0025 and p <0.0001, respectively) and near significance in the P. vivax cases (p = 0.0697; Fig 4D).

All cytokine data were analysed for gender and age influences, but no significant relations were found that would influence the above data. The apparent correlation between age and

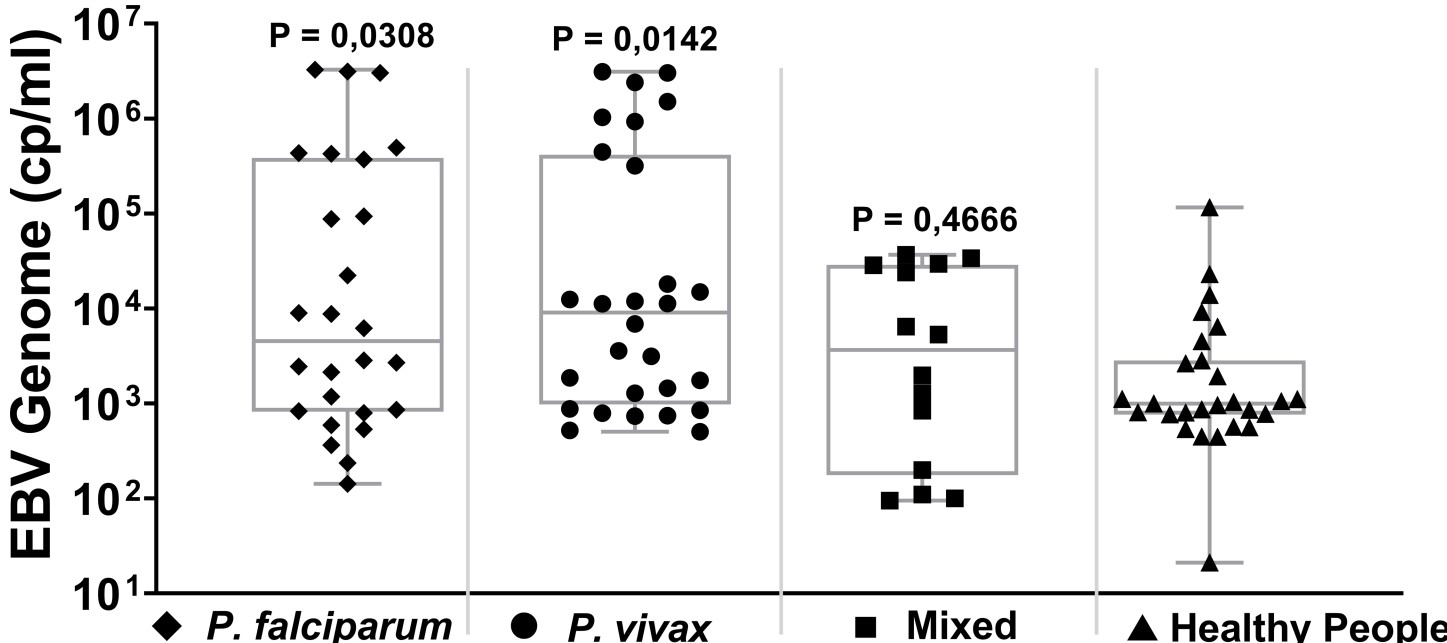

**Fig 2. Quantitative EBV-DNA genome levels (copies/ml) in blood plasma in 3 groups of Malaria cases (*P. falciparum*, *P. vivax* and Mixed) and regional controls.** Box-limits represent the 95% confidence interval and the line represents the median level of EBV-DNA. Statistical analysis was done by one-Way ANOVA (cases versus controls).

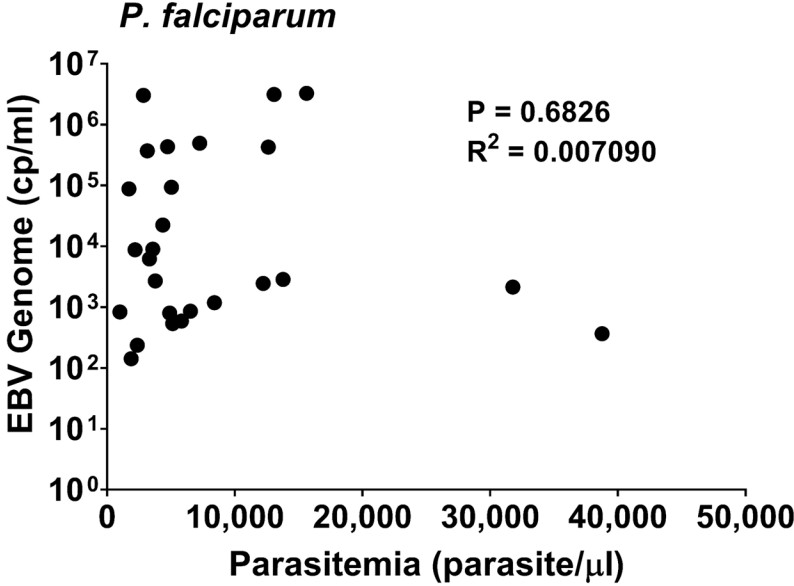

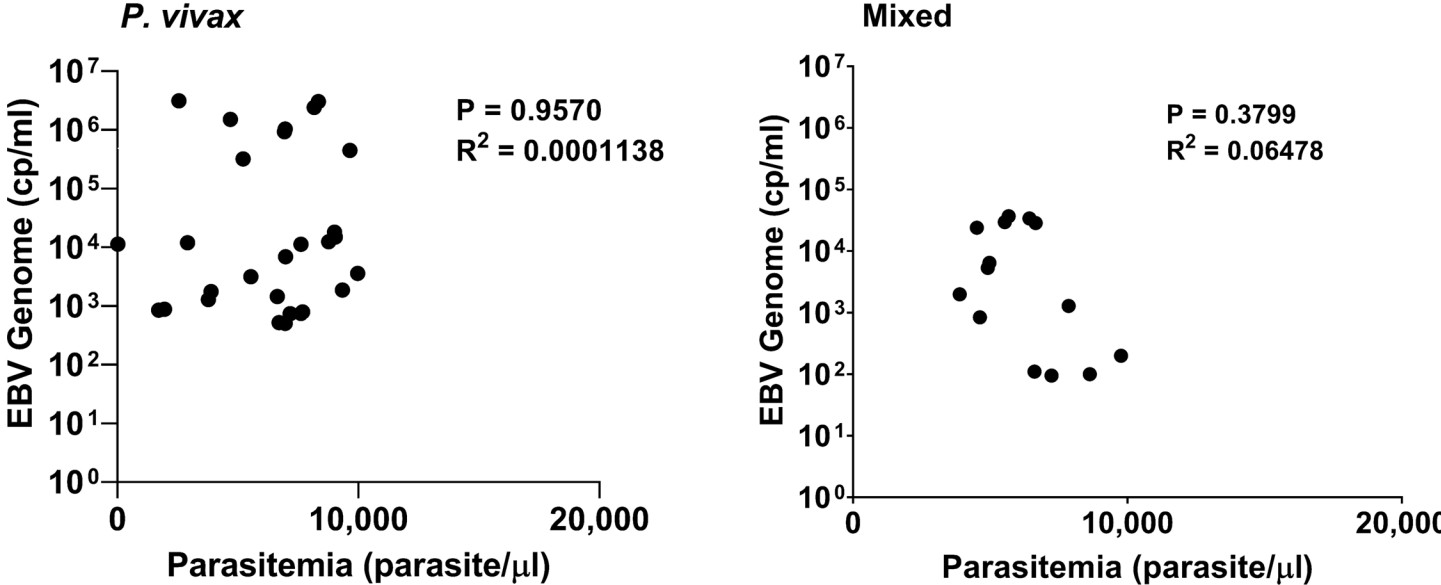

**Fig 3.** Comparison of Parasitemia versus EBV genome levels in Malaria cases; (a) *P. falciparum*, (b) *P. vivax*, (c) Mixed. Statistical analysis ($R^2$) was done by linear regression.

cytokine levels in females with mixed Malaria infection is due to the small number of cases (N = 4) and wide spread of datapoints (S4 Fig).

## EBV genome versus cytokine levels

To study the potential influence of Malaria-induced altered cytokine responses on EBV homeostasis we further analysed the relation between plasma cytokine levels and EBV-DNA load in Malaria cases and controls (Fig 5A–5D). We observed a clear correlation between TNF-α levels and EBV-DNA load in Malaria cases involving *P. falciparum* ($R^2 = 0.8915$), but

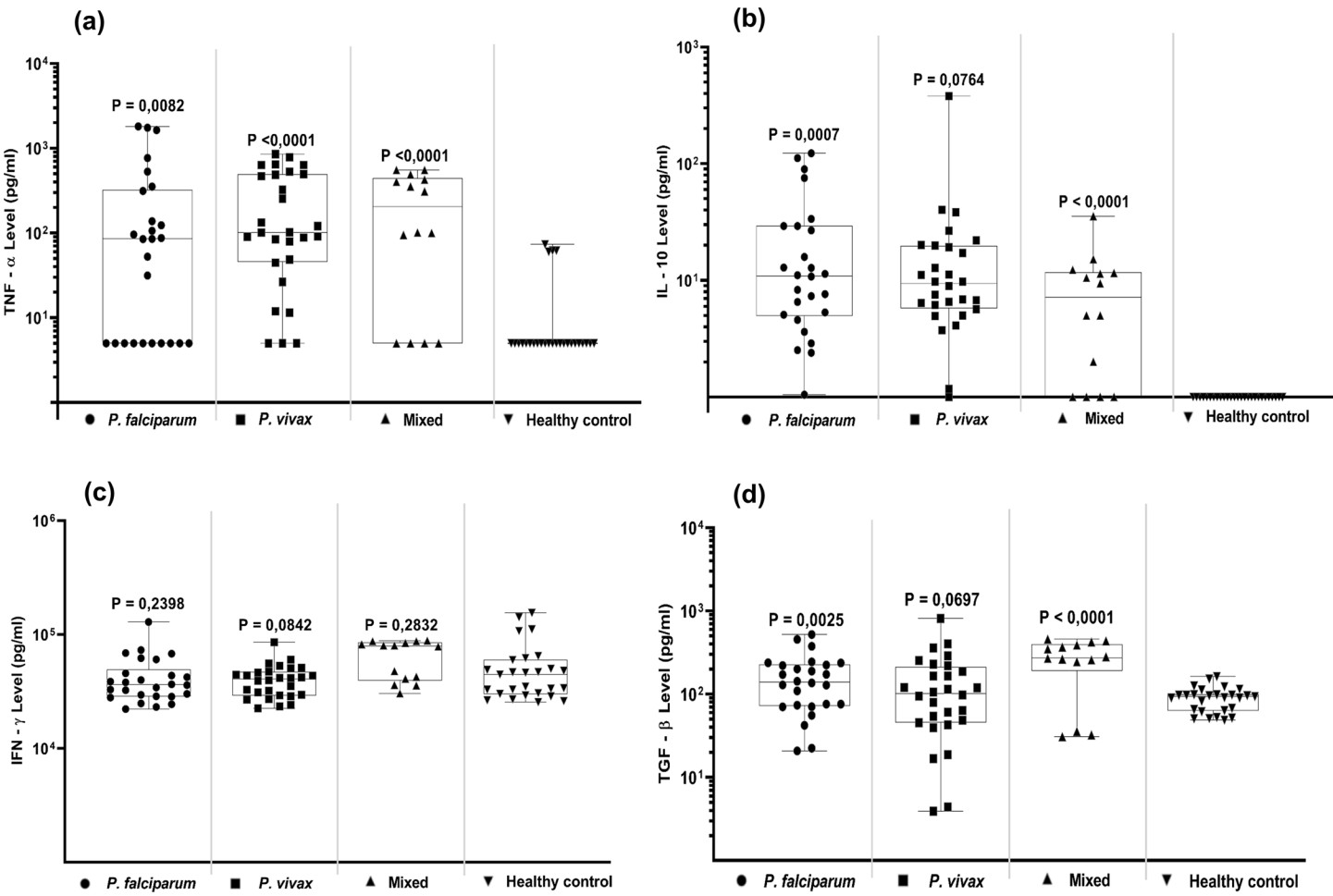

**Fig 4.** Quantitative cytokine levels in Malaria cases and controls; (a) TNF-α, (b) IL-10, (c) IFN-γ and (d) TGF-β. Statistical analysis was done by One-Way ANOVA for cases versus controls.

not in the *P. vivax* or mixed Malaria cases ($R^2$ = 0.08050 and $R^2$ = 0.1137, respectively) nor in the healthy controls ($R^2$ = 0.007915). The levels of other cytokines did not show any correlation ($R^2 <0.3$) with EBV-DNA load in either Malaria or control samples.

## Discussion

All areas of eastern Indonesia including Papua, East Nusa Tenggara, and Maluku Islands are still at high risk of Malaria, whereas subclinical Malaria may circulate in 9.4% of East-Java local people as well [10, 15, 26]. Malaria patients in this study originated from Sumba Island, part of East Nusa Tenggara, whereas controls were from Surabaya, the capital of East-Java. Malaria cases presented with characteristic symptoms and showed high-parasitemia by on-site blood-smear microscopy (Fig 1). Malaria was confirmed by a rapid *Plasmodium* antigen test and expert analysis identified three types of infections, namely *P. falciparum*, *P. vivax* or mixed. EBV infection in Indonesia occurs at early age, establishing a lifelong persistent infection in B lymphocytes in nearly 100% of individuals. EBV infection is generally benign but has a high risk for reactivation and contributes to more serious diseases including cancer [5, 8, 16]. Studies from sub-Sahara Africa have revealed that severe Malaria can trigger the onset of BL, which is preceded by a dysregulated immune balance and an EBV-driven oncogenic event in B-cells

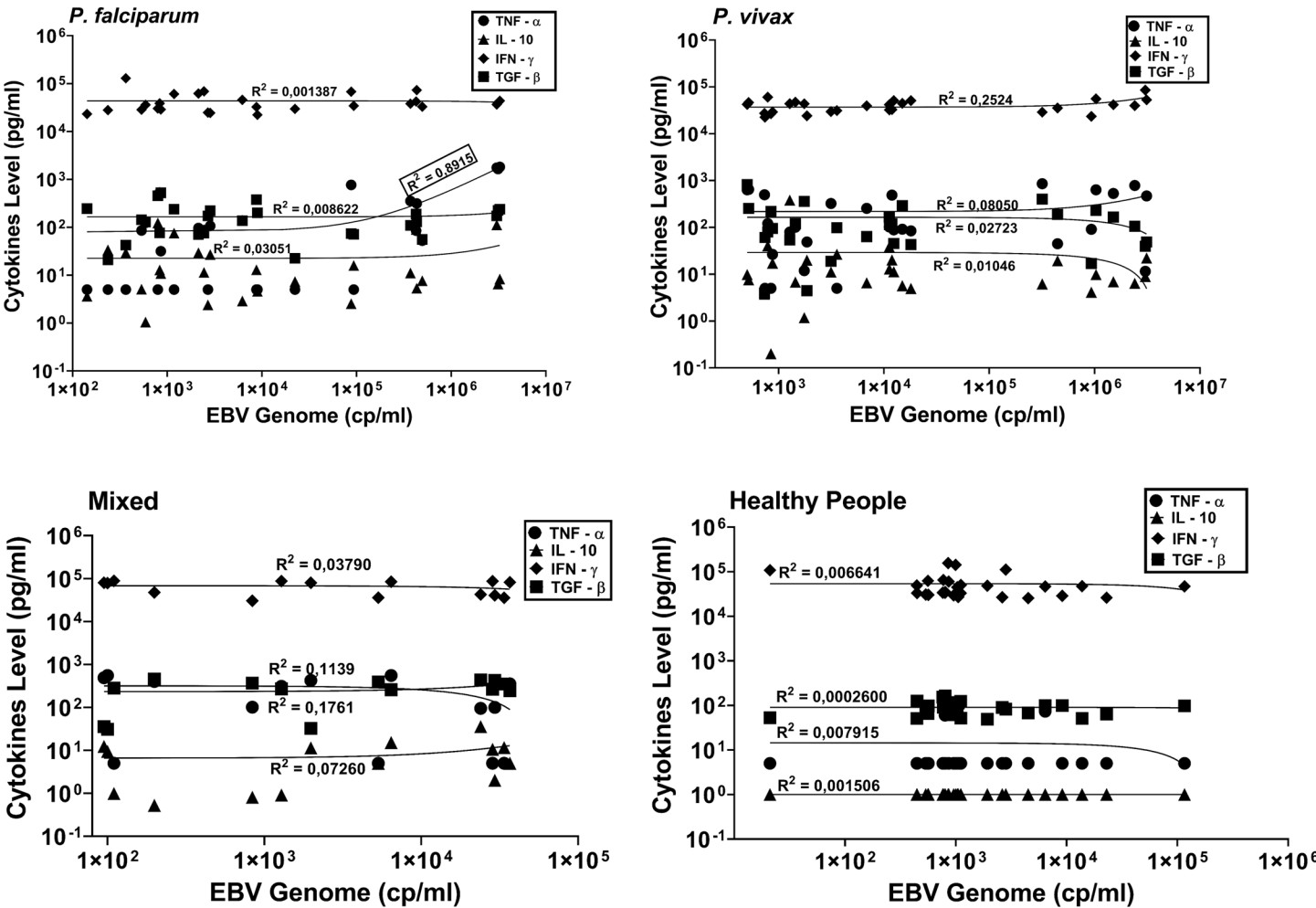

**Fig 5.** Comparison of circulating EBV genome levels and cytokine levels in (a) *P. falciparum* cases, (b) *P. vivax* cases, (c) Mixed cases and (d) healthy controls. Statistical analysis was done by linear regression.

involving chromosomal translocation of cMYC, a key event in BL formation [8–12, 14, 21–23, 27–29]. Although BL has been described in Malaria high-incidence regions in the Indonesian archipelago, such as Papua New Guinea [10], little is known about prevalence and interactions between Malaria and EBV infection in other parts of Indonesia. Most studies on interaction between EBV and Malaria parasites have focussed on *P. falciparum* infections, whereas *P. vivax* and EBV is much less studied [27–30].

It is well known that symptomatic Malaria associates with cytokine disbalances, leading to suboptimal immunity and triggering EBV reactivation from latency [11, 12, 19–23, 27–29]. Importantly, Malaria (especially, involving infection with *P. falciparum*) is associated with increased risk for B-cell malignancies, even in non-endemic countries, although a role of EBV was not always investigated [8, 9, 13, 31]. However, to our knowledge no reports have described Malaria-associated cytokine responses in combination with markers of EBV reactivation, such as circulating EBV-DNA.

In this research, we noted that circulating EBV DNA levels in apparently healthy Surabaya (East-Java) controls (Mean = 7,2 x$10^3$ genome copies/ml) were higher and more variable than previously described for Indonesia, i.c. Jakarta hospital-staff (Mean = 5 x $10^3$ DNA copies/ml)

or Yogyakarta healthy blood donors (Mean = 3.5x $10^3$ copies/ml), which all were elevated compared to Dutch healthy blood donors ($< 2$ x$10^3$ copies/ml) [32–34]. This is in line with previous observations in Kenya, where otherwise healthy people living in malaria-endemic region (Kisumu) had higher EBV-DNA load compared to healthy US-based controls [35]. This observation might reflect general life conditions, being more strenuous in East-Java and Sub-Sahara Africa, or relate to presence of subclinical Malaria, both potentially affecting EBV homeostasis [8, 17, 26]. Importantly, stress-induced corticoid hormones as well as DNA damaging agents/conditions are known to induce EBV lytic cycle and disturb EBV homeostasis [2, 18, 36, 37]. Interestingly, the production/consumption of nitrosamine-containing salty fish recently increased in East-Java during prolonged periods of dry season [17, 38]. Therefore, it may be not surprising that the levels of plasma EBV-DNA in the Surabaya controls are rather high, with occasional individuals experiencing clear EBV reactivation [28, 35]. However, even control subjects with highest EBV-DNA levels did not have any apparent symptoms. These elevated EBV-DNA levels in Indonesian control subjects suggest subclinical EBV reactivation may be common due to local lifestyle, environmental and physical conditions, as previously found in malaria-endemic regions in Africa [26, 35].

Little data are available on EBV-DNA loads during acute Malaria infection in adults. One study from Kenya showed increased EBV-DNA levels associated with acute Malaria in pregnant women [22], whereas most studies addressed Malaria and EBV in African young children at risk of developing BL [14, 21, 27–29]. In our study, we observed significantly elevated mean EBV-DNA loads in Malaria cases compared to regional controls (Fig 2), -despite increased EBV-DNA levels in these controls-, but no relation was found between type of parasite or level of parasitemia and the amount of circulating EBV-DNA in plasma (Fig 3). The elevated EBV-DNA levels were observed both in male and female Malaria cases but did not relate to age (S3 Fig).

Acute malaria infection is associated with a range of pro- and anti-inflammatory cytokine responses [19, 20, 39]. TNF-α and IFN-γ are pro-inflammatory cytokines that stimulate the production of Nitric Oxide (NO) by macrophages and relate to T-cell responses mediating parasite clearance, but also relate to the severity of symptoms. On the other hand, anti-inflammatory cytokines, like TGF-β and IL-10 are related to cell and tissue repair and establishing immune homeostasis [39, 40]. In the present study (Fig 4), we found that healthy controls had very low IL-10 levels, whereas Malaria cases had significant higher levels of IL-10. Low IL-10 levels associate with high stress (emotional or physiological) which may be apply to the present study population (see arguments above) [41]. Elevated IL-10 levels were observed in the Malaria cases, irrespective of subtype, which may reflect reduced T cell responses and increased B-cell activity, with IL-10 serving as endocrine growth factor [42, 43]. TGF-β is associated with repair responses and modifies B-lymphocytes into immunoglobulin-secreting cells and T lymphocytes into cytokine-producing helper cells, thus dampening the creation of cytotoxic T effector cells and natural killer cells [44]. High TGF-β levels correlate with less pathological conditions of Malaria, despite higher parasitemia level [45]. In this research, we found that Malaria cases, irrespective of the subtype, had a broad range of TGF-β responses, but the mean level did not differ significantly from the healthy controls. IFN-γ is considered to control Malaria disease in the early infection blood stage [19, 20]. Elevated levels of circulating IFN-γ are observed in depressed people and persons using anti-depressant drugs, suggesting IFN-γ plays a role in stress-related neuropathology [46]. In this study we found that healthy controls had elevated IFN-γ levels, possibly reflecting stress-health influences in Surabaya. Malaria patients had similar elevated IFN-γ levels as observed in the controls, ruling-out possible parasite-related abnormalities in the IFN-γ response. Above average of TNF-α levels are linked to high deaths rate in children [47, 48]. Disproportionate TNF-α production relates to the

severity of Malaria and may serve as a prognostic factor [49]. Our study seems to confirm these findings, since we observed significantly elevated mean TNF-α levels in all Malaria cases compared to the healthy controls, irrespective of the type of *Plasmodium* infection, age or gender (Figs 4A and S4). Although the levels of TNF-α varied considerably between individual cases, both male and female cases showed increased mean TNF-α levels. The only clear correlation between Malaria-related inflammatory cytokine levels and EBV reactivation, as reflected by increased EBV-DNA levels, was found for TNF-α ($R^2$ = 0.8915) in persons infected with *P. falciparum*, but not in *P. vivax* or mixed-infection cases (Fig 5). This confirms the special interaction between *P. falciparum* and EBV in memory B-cells, as revealed in recent molecular studies [11, 12, 21, 29, 31]. These studies have shown that the *P. falciparum* membrane protein PfEMP1 directly triggers EBV-carrying memory B-cells to establish a pro-inflammatory response and induces B-cell activation/maturation associating with genome-editing activities, thus creating a risk for malignant outcome. EBV-infected B-cells may produce TNF-α directly and high levels of TNF-α are associated with more severe Malaria [47–51]. However, whether EBV and TNF-α together or independently form the driving force in Malaria disease severity (in Indonesia) remains to be studied further. Thus, elevated TNF-α levels may serve as specific indicator for aberrant inflammatory responses in Malaria patients, particularly in individuals infected with *P. falciparum*, that have increased risk for deregulated EBV homeostasis and subsequent risk of developing EBV-related disease [12, 21, 27, 28, 52]. Such inflammation-related deregulated EBV immune balance is reflected not only by elevated EBV-DNA loads in plasma but also by aberrant anti-EBV antibody responses in the co-infected host [14, 53–55].

## Conclusion

This cross-sectional study in a Malaria-endemic region in Indonesia reveals that Malaria parasite co-infection dysregulates the immune system, associating with increased EBV-DNA levels in the circulation, indicative of EBV reactivation. The elevated levels of IL-10, IFN-γ and TGF-β, being not significantly different from regional controls indicates an already pre-existing immune dysregulation due to environmental influences in this Indonesian population, which may affect the normally well-balanced EBV latent carriership. The high levels of pro-inflammatory TNF-α, correlating with increasing circulating EBV-DNA loads especially in *P. falciparum* infected cases, suggests that *P. falciparum* Malaria co-infection causes a impairment of the immune system resulting in systemic reactivation of EBV with potential pathological consequences. Further virological and cancer-registry studies are needed in this population to analyse the suggested association between *P. falciparum* Malaria, EBV and TNF-α in causing chronic or malignant EBV-related disease.

## Supporting information

**S1 Fig. Questionnaire-registered symptoms of Malaria cases caused by *P. falciparum*, *P. vivax* and mixed infection.**
(TIF)

**S2 Fig. EBV-DNA loads in males versus females in Malaria cases and controls.** Box-plot shows 95% confidence interval and median levels and differences between male and female groups were defined by One-Way ANOVA.
(TIF)

**S3 Fig. Overall comparison of EBV DNA levels in Malaria cases versus age.** Statistical analysis was done by linear regression.
(TIF)

**S4 Fig. Comparison of EBV-DNA load and levels of 4 cytokines versus age in male and female cases of Malaria due to infection with *P. falciparum*, *P. vivax* or both.** Statistical analysis was done by linear regression.
(TIF)

## Acknowledgments

We would like to thank: Heny Arwati Ph.D., Dept. Parasitology; Dr. Budi Utomo, Dept. Public Health Sciences-Preventive Medicine; Prof. Indah S. Tantular, Dept. Parasitology and Prof. Maria L. I. Lusida, Institute of Tropical Disease, at Universitas Airlangga, Indonesia for their insight and expertise.

## Author Contributions

**Conceptualization:** Insani Budiningsih, Yoes Prijatna Dachlan, Jaap Michiel Middeldorp.

**Data curation:** Insani Budiningsih.

**Formal analysis:** Insani Budiningsih.

**Funding acquisition:** Usman Hadi, Jaap Michiel Middeldorp.

**Investigation:** Insani Budiningsih, Yoes Prijatna Dachlan.

**Methodology:** Insani Budiningsih, Yoes Prijatna Dachlan, Jaap Michiel Middeldorp.

**Project administration:** Insani Budiningsih, Yoes Prijatna Dachlan, Usman Hadi.

**Resources:** Insani Budiningsih, Usman Hadi.

**Supervision:** Yoes Prijatna Dachlan, Usman Hadi, Jaap Michiel Middeldorp.

**Visualization:** Insani Budiningsih, Jaap Michiel Middeldorp.

**Writing – original draft:** Insani Budiningsih, Jaap Michiel Middeldorp.

**Writing – review & editing:** Yoes Prijatna Dachlan, Usman Hadi, Jaap Michiel Middeldorp.

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
