## [Decision Letter · Decision Letter 0]

15 Jul 2021

PONE-D-21-19672

Quantitative cytokine level of TNF-α, IFN-γ, IL-10, TGF-β and circulating Epstein-Barr Virus DNA load in adults with acute Malaria due to P. falciparum or P. vivax infection or both in a Malaria endemic region in Indonesia.

PLOS ONE

Dear Dr. Middeldorp,

Thank you for submitting your manuscript to PLoS ONE. After careful consideration, we felt that your manuscript requires revision, following which it can possibly be reconsidered. Although your manuscript was of interest to the reviewers, major concerns were related to study design and methods. According to the reviewer # 2 – who has expertise in this field of investigation – it is unlikely that all participants had viral loads above 2.5 log copies and that none of them were negative, not even in the healthy control group. Consequently, it would be helpful that results could be independently validated to demonstrate no cross-contamination with the positive control DNA used for the RT-PCR assay. According to the reviewer #1, the data analysis needs to be revised (for example, analysis should be adjusted for age and sex).  In addition, a significant number of issues should be clarified and/or adjust otherwise the MS’s results may be compromised. For your guidance, a copy of the reviewers' comments was included below.

We look forward to receiving your revised manuscript.

Kind regards,

Luzia Helena Carvalho, Ph.D.

Academic Editor

PLOS ONE

Journal Requirements:

2. "In your Data Availability statement, you have not specified where the minimal data set underlying the results described in your manuscript can be found. PLOS defines a study's minimal data set as the underlying data used to reach the conclusions drawn in the manuscript and any additional data required to replicate the reported study findings in their entirety. All PLOS journals require that the minimal data set be made fully available. For more information about our data policy, please see http://journals.plos.org/plosone/s/data-availability.

Reviewers' comments:

Reviewer's Responses to Questions

**Comments to the Author**

1. Is the manuscript technically sound, and do the data support the conclusions?

Reviewer #1: Yes

Reviewer #2: No

2. Has the statistical analysis been performed appropriately and rigorously? 

Reviewer #1: Yes

Reviewer #2: Yes

3. Have the authors made all data underlying the findings in their manuscript fully available?

Reviewer #1: Yes

Reviewer #2: Yes

4. Is the manuscript presented in an intelligible fashion and written in standard English?

Reviewer #1: Yes

Reviewer #2: Yes

5. Review Comments to the Author

Reviewer #1: Overall Impression: Although the study did not find a clear link between inflammation-associated EBV viral control and severity of malaria infection, the laboratory methods used were robust, and the study offers a contribution to the literature on co-infections in the context of EBV-associated cancer risk.

Introduction: The introduction as written is clear but quite long. At least half of the material in the inflammatory cytokines paragraph could be omitted or transferred to the Discussion. The length of the Background detracts a bit from getting to the hypothesis / objective of the study.

Methods: All of the analyses as presented are not adjusted for age or sex. These two factors should be included, even if in simple linear regression models, to validate that any reported correlations are not driven by a differential distribution of age and / or sex in patients with elevated EBV replication.

Results:

(1) As the reported association between EBV and TNF-alpha is specific to P. falciparum, this needs to be clearly stated everywhere in the Abstract, Introduction, etc…. where it is stated that EBV and malaria-related inflammatory cytokine changes are associated.

(2) Please include the correlations between EBV and TNF-alpha and the other malaria groupings in the text on page 9.

(3) Per Figure 5, it appears that the reported correlation specific to P. falciparum is entirely do to a somewhat linear association at very low levels of the EBV genome number (i.e., the leftmost section of the graph). This should be at least mentioned in the Discussion text.

Discussion: Again, the text here, particularly pages 11-14, is far too much. Please select which paragraphs to highlight based on your findings. An extensive literature review of EBV in different populations and various cytokines that showed no association with EBV are not needed.

Reviewer #2: This manuscript describes a cross-sectional study comparing pro-inflammatory plasma cytokine levels with EBV loads in adults with acute malaria compared to controls residing in Indonesia. It is novel to compared Pf and Pv single and mixed infections within this context since the question often arises as to the specificity of the Pf-EBV interaction. However, without independent confirmation that all the viral loads measured where remarkably high (with none being negative), it is difficult to interpret these findings and their comparisons with cytokine levels.

Major comments:

1. It is extremely unlikely that all your study participants had viral loads above 2.5 log copies and that none of them were negative, not even in your healthy control group. There are many published studies on EBV detection in healthy sero-positive adults that strongly refute your data. Therefore, it would be helpful if your results could be independently validated to demonstrate that there was no cross-contamination with the positive control DNA used for the RT-PCR assay. In addition, the DNA isolated from plasma could be treated with DNAse to determine the fraction of encapsidated virus (Mulama et al IJC 2014) to confirm lytic cycle reactivation in your study populations.

2. The authors state that their study is the first to “measure EBV loads in adults during acute malaria” however they overlooked earlier studies published in Kenya where there was a malaria-exposed adult control group (Moormann et al JID 2005) and other studies by Rochford that measured EBV loads in pregnant women. Therefore, this statement should be tempered. Replications studies done in another study population or at another time that either confirm or refute earlier findings are valid study designs and contribute knowledge to the field.

3. The authors appear to be citing more recent publications or secondary analyzes, as opposed to seminal studies. A more in-depth literature review should be done so this can be remedied.

4. Clinical symptoms listed by individual in Table 1 could be summarized in aggregate across groups, with associated p-values to highlight significant differences, if they exist.

5. The authors interpret higher EBV loads overall in their study groups as being due to stress. Is there any published findings to support this level of stress in people living in Indonesia compared to other study populations residing in malaria endemic regions to support this claim?

6. PLOS authors have the option to publish the peer review history of their article (what does this mean?). If published, this will include your full peer review and any attached files.

Reviewer #1: No

Reviewer #2: No

---

## [Author Response · Author response to Decision Letter 0]

8 Sep 2021

Reply to the reviewer’s comments on manuscript PONE-D-21-19672

We are grateful to the positive and supportive comments of the reviewers.

Our answers to the reviewer’s questions are detailed here below and we have adjusted the revised manuscript accordingly.

Reviewer #1: We thank the reviewer for indicating the positive overall impression.

As suggested, we shortened the introduction by reducing (too many) details on inflammatory cytokines and focus better on the hypothesis / objectives of the study.

We made a statement in the results section that the distribution of EBV-DNA load and 4 individual cytokine levels was analysed for age/sex influences in patients and controls. Data on EBV-DNA male/female comparison are shown in Supplemental. Figure 2 (Box-plot with One-Way ANOVA analysis for DNA-load versus gender in each group).

Data on linear regression analysis for each parameter with age is shown in Suppl. Figure 3 (DNA loads in all Malaria cases) and Supplemental Figure 4 (EBV-DNA and 4 cytokines versus age in P. falciparum, P. vivax and Mixed infections and controls, each separately analysed for male/female status). These analyses did not influence the original results or conclusions. 

#1: We thank the reviewer for this good suggestion and have emphasized the association between EBV and TNF-alpha as being specific to P. falciparum everywhere in the manuscript (abstract, results and discussion).

#2: We have included the correlations between EBV and TNF-alpha for the individual groups in the text of result section as suggested.

#3: The reported correlation specific to P. falciparum is linear and significant (P=0.8915) and holds in both male and female Malaria cases. Due to a systematic error in our DNA load calculation (see reply to reviewer 20 the EBV-DNA loads are now 6-fold lower that initially reported (modified Figure 5).

# Discussion: We have significantly (500 words) reduced the discussion section by removing detailed background information on role(s) of cytokines in anti-Malaria defenses and likewise reduced the number of references. 

Reviewer #2: We thank the reviewer for the positive remarks on the novelty of our findings. We have addressed the concerns on EBV-load measurement.

#1. The EBV-DNA PCR used in this study is an UNG-controlled quantitative PCR with internal calibration and H2O control for each PCR run. Erroneous amplification or PCR cross-contamination is eliminated by the UNG-procedure. Therefore, we consider it highly unlikely that the data are false-positive. This is indicated in the method section.

Unfortunately, due to the Coronavirus epidemic, the Indonesian government has officially called all research PCR instruments in the country to be used for Covid-19 testing only, and we cannot independently re-test our samples. Despite several requests and due to government blocking of new PCR kit import, we are unable to do independent PCR analysis. 

By re-analysing our raw data, we found a systematic error in the EBV-DNA load calculation for all samples (the 10ul PCR input was not re-calculated properly to the 60ul eluate from the Qiagen column). This has resulted in a 6-fold reduction of the reported EBV-DNA levels. All data and figures were corrected for this systematic error.

Yet the resulting EBV-DNA levels remain rather high for healthy individuals, but now they are in line with previously reported data for healthy adults and children in malaria-risk regions in Kenia (ref. 35; Moormann et al., J. Infect. Dis. 2005). One healthy donor had near negative values and her husband had approx. 1100 copies/ml plasma. He has a known IM history > 10 years ago, with persistent high EBV-DNA load in multiple independent PCR analyses but remains without any symptoms. 

Indeed, we were also surprised by elevated EBV-DNA loads found in the nine healthy controls. However, we consider this to be a reflection of the local living conditions, which may be similar to sub-Saharan Africa where healthy children and adults also have higher EBV-DNA loads (ref 35 added). Similar strenuous life conditions in Surabaya, East-Java may affect EBV homeostasis (ref 17, 18, 38). We also noted a recent publication on frequent (9.4%) subclinical Malaria infections (detected by PCR) in the Surabaya urban population (East-Java, Indonesia), where we collected our apparent healthy controls and added this as ref.26. Subclinical malaria may explain -in part- the elevated EBV-DNA levels, similar to malaria-endemic regions in Africa.

We present these arguments (see discussion) about environmental and population stressors which are especially abundant in Surabaya, East-Java compared to other parts of Indonesia (ref 17, 26, 38), and suggest that these may be responsible for the high EBV-DNA loads in this East-Java population, compared to our prior findings in Jakarta and Yogyakarta controls (non-endemic cities in West and central Java). In African healthy controls (adults and children in Kisumu-region, Kenia) the reported EBV-DNA loads are higher in a significant part of the controls compared to US healthy controls (ref. 35) and similar life-stress factors may be due to this.

Although we did not pursue the DNAse treatment, we consider that “overall” the EBV-DNA load in Surabaya EBV carriers is high, whatever the origin (cellular, apoptotic or virion) and is comparable with EBV-DNA loads in healthy African children in the Moormann 2005 study. The environmental factors that may lead to higher EBV-DNA loads (as indicated above), may influence both latent B-cell proliferation (malaria, butyrates, etc) and lytic reactivation (nitrosamines, stress hormones). It is not the aim of this study to analyse the origin of EBV-DNA in detail.

#2. We appreciate the reviewer’s comment and studied the Moormann et al., JID2005 paper. We noticed that this paper deals with children with Malaria and used local non-Malaria children and adult as controls, all being healthy and living locally in a high malaria risk region (Kisumu). Of relevance is their finding that the healthy African children displayed a wide range of EBV-DNA loads and healthy African adults in Kisumu also have elevated EBV-DNA loads compared to US-based individuals. (See our reply here above).

We eliminated the remark that “our study is ‘first’ to measure EBV loads in adults with acute malaria”. 

#3. We appreciate the reviewer’s point of view. Because recent papers cite and discuss earlier seminal findings, we chose to focus on recent publications relevant to our study.

#4. We have aggregated the clinical symptoms in a graphic display (Figure supplement 1) to accommodate the reviewers suggestion. 

#5. Unfortunately, no prior study has addressed this “stress” issue in Indonesia, but it is well known that food and lifestyle (nitrosamines, butyrates, phorbol esters, smoking) as well as stress hormones (glucocorticoids) are inducers of EBV reactivation and contribute to elevated EBV-DNA loads (ref. 18, 37). These factors are abundant in the East-Indonesian region, as mentioned in the discussion with related references (ref.17, 26, 38).

---

## [Decision Letter · Decision Letter 1]

18 Oct 2021

PONE-D-21-19672R1Quantitative cytokine level of TNF-α, IFN-γ, IL-10, TGF-β and circulating Epstein-Barr Virus DNA load in individuals with acute Malaria due to P. falciparum, P. vivax or double infection in a Malaria endemic region in Indonesia.PLOS ONE

Dear Dr.  Middeldorp,

Thank you for submitting your manuscript to PLoS ONE. After careful consideration, we feel that your manuscript will likely be suitable for publication if the authors revise it to address specific points raised now by the reviewers.

According to the reviewers, there are some specific areas where further improvements would be of substantial benefit to the readers (please see reviewer #3).   For your guidance, a copy of the reviewers' comments was included below.  

We look forward to receiving your revised manuscript.

Kind regards,

Luzia Helena Carvalho, Ph.D.

Academic Editor

PLOS ONE

Reviewers' comments:

Reviewer's Responses to Questions

**Comments to the Author**

1. If the authors have adequately addressed your comments raised in a previous round of review and you feel that this manuscript is now acceptable for publication, you may indicate that here to bypass the “Comments to the Author” section, enter your conflict of interest statement in the “Confidential to Editor” section, and submit your "Accept" recommendation.

Reviewer #1: All comments have been addressed

Reviewer #3: (No Response)

Reviewer #4: All comments have been addressed

2. Is the manuscript technically sound, and do the data support the conclusions?

Reviewer #1: Yes

Reviewer #3: Partly

Reviewer #4: Partly

3. Has the statistical analysis been performed appropriately and rigorously? 

Reviewer #1: Yes

Reviewer #3: No

Reviewer #4: Yes

4. Have the authors made all data underlying the findings in their manuscript fully available?

Reviewer #1: Yes

Reviewer #3: Yes

Reviewer #4: Yes

5. Is the manuscript presented in an intelligible fashion and written in standard English?

Reviewer #1: Yes

Reviewer #3: Yes

Reviewer #4: No

6. Review Comments to the Author

Reviewer #1: The authors’ responses have adequately addressed all of my concerns, and I recommend publication of the current version of the manuscript revision.

Reviewer #3: Quantitative cytokine level of TNF-α, IFN-γ, IL-10, TGF-β and circulating Epstein-Barr Virus DNA load in individuals with acute Malaria due to P. falciparum, P. vivax or double infection in a Malaria endemic region in Indonesia by Budiningsih et al

The authors report their findings of correlations between EBV level and parasitemia in in subjects in Indonesia. This works sheds light on the interaction between malaria parasites and EBV in a population with low incidence of Burkitt lymphoma. The authors are to be congratulated for designing and implementing this study. It is timely and represents an important step forward for understanding the pathobiology of EBV and malaria. Their manuscript would benefit from some improvements as suggested below:

General comments:

a) The authors report the study as a case-control study, but it is not clear whether this is not a cross-sectional study. This difference is important and needs further clarification. A cross-sectional study is one where all assessments (exposure and outcome) are made at the same point in time. In this case, if classification as either a case or control is made based on responses to a questionnaire and blood test made at the same time on the subject, the study should be classified as a cross-sectional study, with subjects enrolled from two different locations. A case control study is defined as enrollment of subjects with or without the condition of interest (in this case malaria). Thus, cases are evaluated for the presence of the condition and excluded if they don’t have the condition, while controls are similar to the cases in all respects except in having the outcome of interest. The requirement for comparability of cases and controls imposes on the author the need to describe the source of cases, the population which they represent, and to describe the controls so readers can conclude whether the controls are from a population where cases originated. These details are lacking or do not seem to be met for a case-control design. I suggest that the authors report this as a cross-sectional study. Regardless, the authors need to report the subjects who were assessed and if any were excluded and the reasons for exclusion.

b) The authors need to clarify what they refer to as malaria. Malaria is the disease ie, infection PLUS symptoms. Individuals with malaria have a range of symptoms categorized as severe, moderate, mild, or asymptomatic. These are defined in a standard way by the WHO and should be used as such. For example, hyper-parasitemia is defined as parasitemia of 250,000/uL. It is doubtful that the subjects included here meet that definition, but if they do they should us the standard WHO definition and show the results in a primary table.

c) The analytic methods need to specify and distinguish the primary hypotheses being tested from the secondary explorative hypotheses being tested. The methods should also specify how multiple comparisons are being handled.

d) The authors need to provide some epidemiological context for their work – for example what is the malaria pressure in heir population - this can be provided in the form of malaria prevalence, entomology inoculation rates, and the age when severe malaria is observed in that population or the annual burden of malaria. This background information would be helpful for readers to understand how the cases being reported compared to the malaria patterns observed in the area.

e) The authors need to provide more information about the sites for the enrollment – what are these malaria hospitals?

f) Please add page numbers!

Specific comments

g) The sentence “Enhanced EBV-DNA levels were significantly more frequent in P. falciparum and P. vivax infections (P<0.05) compared to controls.” In the abstract and elsewhere in the manuscript is confusing. Levels are higher, positivity is more frequent. The authors need to read the manuscript carefully and distinguish instances where they are talking about EBV DNA positivity as a binary variable versus when they are talking about the quantity.

h) 2nd paragraph of the introduction – please add some malaria contextual information as suggested above. Some of this information is I the first 2-3 paragraphs of the discussion.

i) Last paragraph of the introduction – please “in Indonesia” where applicable and “in East Africa” or “Kenya’ as applicable, eg, “The aim of this research was to investigate how acute Malaria dysregulates EBV homeostasis and what cytokines would be involved “in Indonesia”

j) In the material and methods revise “Malaria samples”… to “venous blood samples”

k) Please provide more information about the sites where the subjects were enrolled – hospitals, communities etc etc using the STROBE guidelines.

l) Please clarify whether all subjects were assessed for malaria – this would include a screening for temperature, symptoms, and parasites. A standard definition of malaria should be used, denoting cases as either severe, moderate, mild, or asymptomatic.

m) For healthy controls, the test used to exclude infection should be stated as well as its limit of detection.

n) The word “Clinical” in the sentence “Clinical diagnosis of Malaria was confirmed by a laboratory test for Malaria antigen detection, i.c. “Rapid Diagnostic Test” (RDT) [CareStartTM Malaria Pf/PAN (HRP2/pLDH) Ag Combo RDT, lot.nr. RMRM-01071, ACCESSBIO, Somerset, NJ, USA].” should be replaced with “presence of plasmodial parasite infection”. Malaria by definition is clinical, ie, associated with symptoms. Thus, the authors could say “the diagnosis of malaria was made based on demonstration of presence of parasites, based on …tests, in patients with a fever and XX symptoms”

o) The sentence “… followed-up with thick smear microscopy by expert parasitologists to confirm the parasite species and to quantify the proportion of infected red blood” needs to be revised for clarity and correctness. It is impossible to determine the number of red blood cells infected from a thick film.

p) Please provide some additional details of the giemsa stain (concentration and duration of staining, and how quickly the slides were made after collecting the blood)

q) In the statistics section – this needs to be revised for clarity “ The correlation between EBV genome and Malaria parasites levels in a population was defined by student t-test” this test is normally used to test for quality of means – is this what is meant here? The authors need to describe what is being evaluate – means, correlation, association and what statistics are being used. This section requires careful attention.

r) In the results section - the definition of hyper-parasitemia is unclear here. Do these patients really have severe malaria? Could other measures of malaria severity be provided? Eg, hemoglobin, platelet count, WBCs?

s) If the authors believe that the EBV levels might differ by type plasmodium – this should be articulated. Are there differences in the severity of malaria? Given the limitations of sample size, this segmentation of data should be considered carefully.

t) The results presented as “Overall, the mean EBV genome level was significantly higher in cases of P. falciparum (mean level = 4.4 x105 copies/ml; SD = 9,9 x105) and P. vivax (mean = 4,6 x105 copies/ml; SD = 9,1 x105) infection compared to the mean level in the healthy controls (mean = 7,2 x103 copies/ml; SD = 2,2 x104) (P < 0.05 for mean levels; Student t-test).” are interesting, but I would suggest re-vising this such that the different statistical results can be presented, ie, “Compared to healthy controls (mean = 7,2 x103 copies/ml; SD = 2,2 x104), EBV levels were significantly elevated in those with P. falciparum (XX, XXX, P=XXX) and those with P. vivax (XX, XX, P=XX).” The authors could also compare the different groups according to log levels, i.e., 2 log higher …

u) I think the authors need to present the significant cytokine results and report those that are not significant as not significant without going into details.

v) The discussion should be shortened and focused on the key findings, what they mean, and the strengths and weaknesses of this study, and brief suggestion of the next steps.

Reviewer #4: Reviewers comments

Manuscript Number: PONE-D-21-19672R1

Manuscript Title: Quantitative cytokine level of TNF-α, IFN-γ, IL-10, TGF-β and circulating Epstein-Barr Virus DNA load in individuals with acute Malaria due to P. falciparum, P. vivax or

double infection in a Malaria endemic region in Indonesia.

Summary of article

The work is very interesting, novel and is well suited for publication in PLOS-ONE.

I have reviewed the author’s response to the previous reviewers comments and am happy to see that all relevant points were taken on board to improve the manuscript.

Introduction

Line – “EBV was first identified in cells of Burkitt Lymphoma (BL), an endemic cancer among sub-Saharan children, that is triggered by co-infection with Malaria parasites [reviewed in 8, 9]. BL is the most common cancer in children living in Malaria endemic regions in sub-Sahara Africa and Papua New Guinea [9,10],” - recent publications have been published highlighting new perspectives such as Ellis et al 2021 (see below) which can be referenced

Ellis, T., Eze, E. & Raimi-Abraham, B.T. Malaria and Cancer: a critical review on the established associations and new perspectives. Infect Agents Cancer 16, 33 (2021). https://doi.org/10.1186/s13027-021-00370-7

Line – “There appeared to be a direct correlation between increases in plasma EBV viral load and progression of endemic Burkitt Lymphoma” abbreviate to BL

Information in the introduction on the four cytokines chosen to quantify in the study should be provided in greater detail beyond the summary paragraph provided.. Further details on the selection/choice of specific cytokines used in this study is important. For example “ in this study the amount of x,x,x, and x plasma cytokines was evaluated. X has a role in xx etc repeat.

Materials and Methods/ Results and Discussion

Sample collection and Malaria parasite analysis – because of the age association with relationship between malaria and EBV it would have been useful for the field to have full malaria patient data i.e. positive parasites of P. falciparum (n=26), P. vivax (n=28), and mixed (P. falciparum and P. vivax) (n=14), or healthy controls (n=27) data presented in a table with age information to know for example how many patients with P. falciparum infection were under 5 etc. This should be reflected in the results and discussion. Whilst the gender could be of interest especially in any of the samples were from pregnant women, the age is of greater interest and should be included.

The authors should highlight any sample data from pregnant women.

Line – use of Kenya and Kenia in the manuscript change all “Kenia” to Kenya for continuity.

7. PLOS authors have the option to publish the peer review history of their article (what does this mean?). If published, this will include your full peer review and any attached files.

Reviewer #1: No

Reviewer #3: No

Reviewer #4: No

---

## [Author Response · Author response to Decision Letter 1]

23 Nov 2021

PONE-D-21-19672R16. 

Authors Reply to: Review Comments to the Author

Reviewer #1: The authors’ responses have adequately addressed all of my concerns, and I recommend publication of the current version of the manuscript revision.

Au reply: We thank the reviewer for the positive consideration.

Reviewer #2: No further comments upon the first reply and revised manuscript.

Au reply: The Reviewer #4 has taken notice of these corrections and consider them adequate (see comments and reply for reviewer #4).

Reviewer #3: The authors are to be congratulated for designing and implementing this study. It is timely and represents an important step forward for understanding the pathobiology of EBV and malaria. Their manuscript would benefit from some improvements as suggested below:

Au reply: We thank reviewer #3 for the congratulations and positive response and the proposed adjustments on which we reply here below:

General comments:

a) The authors report the study as a case-control study, but it is not clear whether this is not a cross-sectional study. This difference is important and needs further clarification. I suggest that the authors report this as a cross-sectional study. Regardless, the authors need to report the subjects who were assessed and if any were excluded and the reasons for exclusion.

Au reply: The authors agree that this research is a cross-sectional study (as now mentioned in the abstract and conclusion section (line 19 and 338, respectively). 

This study has two specific goals: (1) to compare EBV genome and cytokine levels in acute Malaria cases infected with different malaria parasites (P. Falciparum or P. vivax or mixed) versus regional healthy controls and (2) to study the relation between defined cytokine levels and EBV-DNA loads. Sampling was done based on inclusion and exclusion criteria as was mentioned in the original sections on Sample collection and Malaria parasite analysis: 

- Inclusion criteria of clinical samples: 

1) People who live in high endemic malaria region, i.c. Sumba Island, Nusa Tenggara province, Indonesia

2) Suffering from clinical malaria (clinical symptoms detailed in Suppl. Table 1) plus having a positive malaria rapid test result

3) Age and gender were not limiting 

- Inclusion criteria of healthy control samples:

1) Healthy residents who live in Surabaya city, a regional low Malaria region

2) Not suffering from malaria or other acute or chronic diseases for 1 year

3. Age: adulthood 

- Exclusion criteria of patient and control samples: 

1) Donors suffering from HIV and/or sexually transmitted diseases

2) Being non-coöperative (not filling-out questionaire or not giving consent)

b) The authors need to clarify what they refer to as malaria. Malaria is the disease ie, infection PLUS symptoms. Individuals with malaria have a range of symptoms categorized as severe, moderate, mild, or asymptomatic. These are defined in a standard way by the WHO and should be used as such. For example, hyper-parasitemia is defined as parasitemia of 250,000/uL It is doubtful that the subjects included here meet that definition, but if they do, they should us the standard WHO definition and show the results in a primary table.

Au reply: All patients had clinical symptoms characteristic of acute Malaria at intake (see original methods section and Supplemental Figure 1) and had a positive on-site score in the Rapid Antigen test (RDT defined in Methods). All patients subsequently visited the local hospital for more detailed examination by an expert parasitologist (Questionaire, HIV and STD testing and blood parasite scoring/typing). 

Parasitemia levels in blood were categorized as 4 groups according to WHO-2010, as specified in the Methods section (ref. 24).

All parasitemia levels of P.falciparum, P. vivax and mixed were >50 parasites per single oil-immersion thick film field (therefore categorized as high parasitemia or Group 4), except one patient with P.vivax. 

The final number of parasites per μL of blood was calculated as the formula: [(counted parasites/500WBC) x counted or assumed WBC/μL(8000)] as prescribed by the WHO-2010 guidelines. 

These details are mentioned in the Methods and shown in Figure 1 (Parasitemia levels) and Suppl. Figure 1 (Symptoms).

We changed hyper-parasitemia into high parasitemia (2x) in the manuscript.

c) The analytic methods need to specify and distinguish the primary hypotheses being tested from the secondary explorative hypotheses being tested. The methods should also specify how multiple comparisons are being handled.

Au reply: The comparison of EBV DNA levels between Malaria cases and controls was done by unpaired t-test, Pearsons R test was used for correlation analysis between EBV-DNA levels and parasite or cytokine levels. Linear regression analysis was used to determine the relationship between age, gender and one or more independent variables (i.c. EBV genome and cytokine levels), as already mentioned in the statistical analysis.

d) The authors need to provide some epidemiological context for their work – for example what is the malaria pressure in their population - this can be provided in the form of malaria prevalence, entomology inoculation rates, and the age when severe malaria is observed in that population or the annual burden of malaria. This background information would be helpful for readers to understand how the cases being reported compared to the malaria patterns observed in the area.

Au reply: The background on Malaria-incidence and spread in this study population (Sumba Island) was deduced from the ongoing Indonesian Malaria reduction campaign, summarized briefly in Lancet Global Health (2018), indicated here as ref. 15. The prevalence of Malaria in the control population (Surabaya, East-Java) was described by Arwati et al., 2018, named here as ref. 26. We indicated some background info on EBV infection in Indonesia on lines 65-70. 

Although many studies on malaria epidemiology, immunology, and drug resistance have been conducted at many sites in Indonesia, there is little published literature describing malaria prevalence at the district, provincial, or national level (ref. 15). 

For extra info on local Malaria-situation in Sumba Island to the reviewer, not included in the manuscript because of ref. 15: Prior research (Syafruddin et al, 2009) investigated two stages of cluster sampling malaria prevalence surveys in the wet season and dry season across West Sumba, Nusa Tenggara Province, Indonesia. The overall prevalence of malaria infection in the West Sumba District was 6.83% (95% CI, 4.40, 9.26) in the wet season and 4.95% (95% CI, 3.01, 6.90) in the dry. In the wet season Plasmodium falciparum accounted for 70% of infections; in the dry season P. falciparum and Plasmodium vivax were present in equal proportion. Malaria prevalence varied substantially across the district; prevalence rates in individual sub-villages ranged from 0-34%. The greatest malaria prevalence was in children and teenagers; the geometric mean parasitemia in infected individuals decreased with age. Malaria infection was clearly associated with decreased hemoglobin concentration in children under 10 years of age, but it is not clear whether this association is causal. Syafruddin D. et al. Seasonal prevalence of malaria in West Sumba district, Indonesia. Malar J. 2009;9,8:8. doi: 10.1186/1475-2875-8-8. PMID: 19134197.

e) The authors need to provide more information about the sites for the enrollment – what are these malaria hospitals?

Au reply: As part of Sumba-Island health care system, experienced health workers visited each of the suspect malaria patients at their homes in different villages and then examined the patients on site with RDT. If the RDT result is positive, the patient was referred to the nearest public health center in the district (such as Public Health Centre Kori and Public Health Melolo on Sumba island) to follow-up with clinical and microscopic examination of the malaria status and to obtain their questionnaire and plasma. The plasma of malaria patients was placed in a cool box with dry ice and shipped to the Institute of Tropical Disease, University of Airlangga, Surabaya. Upon arrival, the plasma was immediately aliquoted and frozen at –80°C. When being used, plasma samples were thawed and stored on melting ice or in a refrigerator at +2°-8°C. 

The manuscript is adjusted with these details (see lines 96-106). 

f) Please add page numbers!

Au reply: We added page numbers and line numbering throughout the manuscript.

Specific comments

g) The sentence “Enhanced EBV-DNA levels were significantly more frequent in P. falciparum and P. vivax infections (P<0.05) compared to controls.” In the abstract and elsewhere in the manuscript is confusing. Levels are higher, positivity is more frequent. The authors need to read the manuscript carefully and distinguish instances where they are talking about EBV DNA positivity as a binary variable versus when they are talking about the quantity.

Au reply: We screened and adjusted the manuscript as suggested

h) 2nd paragraph of the introduction – please add some malaria contextual information as suggested above. Some of this information is in the first 2-3 paragraphs of the discussion.

Au reply: See Au reply under d). 

In Indonesia, with around 270 million population that is 100% positive for EBV, about 10,7 million people are still living in Malaria endemic areas [15]. Children in Indonesia are exposed to EBV at early age with high dose of EBV via saliva (pre-chewed food). Later in life chronic exposure to EBV carcinogens, such as formalin, tobacco additives, herbal drugs/oils, butyrate acid (dried meat) and nitrosamine (dried salty fish) are common in Indonesia, which can trigger aberrant and pathogenic EBV activity and malignancy [16-18].

More detailed background was excluded from the introduction to reduce the manuscript size upon earlier reviewer’s suggestion (R1) and is now summarized in the discussion only. Because we found a rather high EBV-DNA load in Surabaya controls compared to our previous studies in Yogyakarta and Jakarta (= new result), we placed information on local life-conditions and malaria incidence in the discussion section (lines 264-282; ref. 17, 26, 35-38).

i) Last paragraph of the introduction – please “in Indonesia” where applicable and “in East Africa” or “Kenya’ as applicable, eg, “The aim of this research was to investigate how acute Malaria dysregulates EBV homeostasis and what cytokines would be involved “in Indonesia”.

Au reply: We adjusted this issue in the manuscript, where applicable.

j) In the material and methods revise “Malaria samples” to “venous blood samples”.

Au reply: We adjusted this in the manuscript

k) Please provide more information about the sites where the subjects were enrolled – hospitals, communities etc etc using the STROBE guidelines

Au reply: We adjusted this issue: see also Au reply to e)

l) Please clarify whether all subjects were assessed for malaria – this would include a screening for temperature, symptoms, and parasites. A standard definition of malaria should be used, denoting cases as either severe, moderate, mild, or asymptomatic.

Au reply: These details are mentioned in the manuscript and follow the WHO-2010 guidelines. 

Clinical samples obtained were from symptomatic malaria patients, who often appear with a spectrum of symptoms such as fever, headaches, nausea, pale, and conjunctival pallor (see Suppl. Figure 1). 

Clinical findings were confirmed on-site by a “Rapid Diagnostic Test” (RDT) for malaria antigen detection (specified in Methods). RDTs are available in the remote areas in all Indonesia provinces, as an alternate way of quickly establishing the diagnosis of malaria infection. 

All positive RDTs were followed-up in the nearest regional hospital on Sumba Island (see above) with thick smear microscopy to confirm the parasite species and to quantify the proportion of infected erythrocytes in relation to a predetermined number of WBCs. After the examinations were concluded and prior to the start of treatment, fresh venous blood was collected for plasma preparation with informed consent from the patients who were confirmed malaria positive.

m) For healthy controls, the test used to exclude infection should be stated as well as its limit of detection.

Au reply: We did not specifically test the health status of the controls. We performed only physical examinations and personal interview to establish their condition at the time of sampling: all were healthy with no underlying disease in the recent year.

n) The word “Clinical” in the sentence “Clinical diagnosis of Malaria was confirmed by a laboratory test for Malaria antigen detection should be replaced with “presence of plasmodial parasite infection”. Malaria by definition is clinical, ie. associated with symptoms. Thus, the authors could say “the diagnosis of malaria was made based on demonstration of presence of parasites, based on …tests, in patients with a fever and XX symptoms”

AU reply: We adjusted the text as suggested (Lines 114-117). “The diagnosis of Malaria was confirmed in symptomatic patients by demonstrating the presence of plasmodial parasite infection in a fresh drop of finger prick blood using a Rapid test for Malaria antigen detection, i.c. “Rapid Diagnostic Test” (RDT), …

o) The sentence “… followed-up with thick smear microscopy by expert parasitologists to confirm the parasite species and to quantify the proportion of infected red blood” needs to be revised for clarity and correctness. It is impossible to determine the number of red blood cells infected from a thick film.

AU reply: We modified the text as suggested, by referring to the WHO-2010 guidelines (ref. 24). See lines 118-122: “All cases with a positive RDT were followed-up in the regional health center(s) on Sumba Island by further examination using fresh venous blood and thick smear microscopy by an expert parasitologists to confirm the parasite species and to quantify the proportion of infected red blood cells, according to WHO guidelines [24]. Briefly,…etc”.

p) Please provide some additional details of the Giemsa stain (concentration and duration of staining, and how quickly the slides were made after collecting the blood).

AU reply: These details are compliant with the WHO-2010 guidelines (24) and are not mentioned in such detail in the manuscript. 

We gladly will specify our procedures for the reviewer here: Using a clean lancet the tip of a finger was punctured and the first drop of blood was wiped away with clean gauze. The next drop of blood was touched with a clean slide and a small drop of blood was placed in the center of the pre-cleaned, labeled slide and was spread in a circular pattern until the size was of a dime (1,5 cm2), the corner of another slide was used to spread the blood drop. The slides were laid flattered and the smears were allowed to air-dry thoroughly in 20-30 minutes with a fresh air fan being protected from heat, dust and insects. 

Fresh working Giemsa stain (2.5% w/v in PBS) was prepared in a staining jar with 40ml fills and 2 drops of Triton X-100 were added. Fixed thick smear slides were placed into Giemsa stain for 45-60 minutes. Thereafter slides were removed and left in PBS for 5 minutes and dried upright in a rack. Staining Procedure for Quality Control = a positive smear (malaria) was included with each new batch of working Giemsa stain. 

q) In the statistics section – this needs to be revised for clarity “The correlation between EBV genome and Malaria parasites levels in a population was defined by student t-test” this test is normally used to test for quality of means – is this what is meant here? The authors need to describe what is being evaluate – means, correlation, association and what statistics are being used. This section requires careful attention. 

AU reply: The statistics section has been modified in detail as suggested by the reviewer #3 (Lines 162-167). Detailed calculation results for the reviewer #3 are shown here below.

EBV Genome = P. falciparum vs Healthy control

EBV Genome = P. vivax vs Healthy control

EBV Genome = Mixed (P. falciparum and P. vivax) vs Healthy control

P. falciparum Parasitemia vs EBV Genome

P. vivax Parasitemia vs EBV Genome

Mixed (P. falciparum and P. vivax) Parasitemia vs EBV Genome

r) In the results section - the definition of hyper-parasitemia is unclear here. Do these patients really have severe malaria? Could other measures of malaria severity be provided? Eg, hemoglobin, platelet count, WBCs?

AU reply: see Au reply to reviewer’s questions under b). 

At intake, all patients had multiple symptoms characteristic of acute Malaria (listed in Suppl. Figure 1), plus a positive “on site” RDT and >50 parasites per thick blood smear (category 4+) by subsequent expert parasitologist examination (WHO-2010 guidelines) prior to treatment. We did not make any statement on the clinical severity of Malaria, other than being acute symptomatic and RDT confirmed Malaria cases. To avoid confusion, we changed hyper-parasitemia into high (4+) parasitemia (2x) in the manuscript and show all parasitemia numbers in Figure 1.

s) If the authors believe that the EBV levels might differ by type plasmodium – this should be articulated. Are there differences in the severity of malaria? Given the limitations of sample size, this segmentation of data should be considered carefully.

AU reply: We indeed found significantly elevated mean levels of EBV-DNA in plasma in both P. falciparum and P.vivax Malaria cases versusthe healthy controls (Mixed cases also had elevated EBV-DNA, but not significantly), as stated in the results, but the mean EBV-DNA levels were not different among both malaria groups. 

We did not address case-to-case differences in Malaria severity other than having multiple symptoms and Group-4+ level of parasitemia in all but 1 case of Mixed parasite infection. Details are shown in Figure 1 (parasitemia level) and in Supplemental Figure 1 (distribution of symptoms).

Sample size for this cross-sectional study was defined using the following formula (Hulley et al., 2013):

N=[(Zα+Zβ)/(0,5 ln⁡((1+r)/(1-r)) )]^2+3 = [(1,96+0,842)/(0,5 ln⁡((1+0,653)/(1-0,653)) )]^2+3=15,872 ≈ 16

If the estimated degree of relationship is a moderate degree with a correlation coefficient (r) = 0,653; standard deviation (type I error rate) = 5%, α = 0,05 (p = 0,05), then Zα = 1,96 (from Z table); standard deviation (type II error rate) = 20% and the standard normal deviate for β = 0,2 (research power 80%), then Zβ = 0,842, thus the minimum sample size is 16 samples. If a 10% drop out is expected (do = 0,1), the sample size with drop out correction is:

 n_do= n/(1-do)^(2 ) = 16/(1-0,1)^(2 ) =19.75≈20

Therefore, a minimum sample size of 20 samples is needed for this research. 

We did not specify this calculation in the methods, but it was part of the initial planning and study approval protocol.

Note: due to the decreasing incidence of Malaria in Sumba Island it was difficult to find equal numbers of mixed infection cases in this cross-sectional survey.

 t) The results presented as “Overall, the mean EBV genome level was significantly higher in cases of P. falciparum (mean level = 4.4 x105 copies/ml; SD = 9,9 x105) and P. vivax (mean = 4,6 x105 copies/ml; SD = 9,1 x105) infection compared to the mean level in the healthy controls (mean = 7,2 x103 copies/ml; SD = 2,2 x104) (P < 0.05 for mean levels; Student t-test).” are interesting, but I would suggest re-vising this such that the different statistical results can be presented, ie, “Compared to healthy controls (mean = 7,2 x103 copies/ml; SD = 2,2 x104), EBV levels were significantly elevated in those with P. falciparum (XX, XXX, P=XXX) and those with P. vivax (XX, XX, P=XX).” The authors could also compare the different groups according to log levels, i.e., 2 log higher …

Au reply: We modified the text as suggested (lines 190-195). “Compared to the healthy controls (mean EBV DNA level = 7,2 x103 copies/ml; SD = 2,2 x104), cases of P. falciparum (4.4 x105 copies/ml; SD = 9,9 x105) and P. vivax infection (4,6 x105 copies/ml; SD = 9,1 x105) had significantly higher mean EBV DNA levels (P = 0.0308 and 0.0142, respectively; Unpaired t-test).”

u) I think the authors need to present the significant cytokine results and report those that are not significant as not significant without going into details.

Au reply: Because -in our opinion- this is a first study detailing 4 cytokine levels versus EBV-DNA load in acute symptomatic Malaria cases, we consider it relevant to mention all result details. In particular, the high levels of pro-inflammatory TNF-α found in this research may reflect how malaria P. falciparum infection causes dysregulation of endogenous EBV co-infection by impairment of the basic immune regulation. The elevated levels of IL-10, IFN-γ and TGF-β, though not significantly different from regional controls indicate a (pre-existing) dysregulation of normal immune balance (due to environmental “life-style” stress in the local Indonesian population), which may also affect EBV balanced latency. This may be similar to observations in healthy controls living in malaria-endemic regions in Africa, as indicated in the discussion (ref. 35).

v) The discussion should be shortened and focused on the key findings, what they mean, and the strengths and weaknesses of this study, and brief suggestion of the next steps.

Au reply: Because our study is first in quantitatively analyzing the levels of 4 different cytokines versus the EBV-DNA loads in acute malaria cases, in particular in Indonesia where such details are not described before, we consider a detailed description and discussion (lines 263-281) of the underlying regional life-style/environmental factors in Indonesia that may affect “normal” cytokine and EBV balances is relevant here and has not been considered as such in previous studies (ref. 35). Such underlying dysregulated EBV latency may be more profoundly altered upon repeated episodes of severe malaria, and especially in young children in endemic areas may increase the risk of lymphoma formation (ref’s 8-12, 14, 21-23, 27-29). 

The strength of our study is the combined marker analysis (4 cytokines and EBV-DNA) in 3 groups of Malaria cases, being Pf, Pv and Mixed. A weakness is the relatively low number of cases overall, particularly in the mixed group.

The key finding in our work is the high level of pro-inflammatory TNF-α correlating with elevated EBV-DNA levels particularly in P. falciparum malaria cases, thereby supporting the previously described special interaction of P. falciparum parasites with EBV-infected B-cells involving the CIDR1α domain of P. falciparum Erythrocyte Membrane Protein 1 (PfEMP1), (ref 11-13). This is highlighted in the abstract, the discussion and conclusion. Further cancer-registry and clinical-immunological studies are needed to analyse this suggested interaction and its consequences in the Indonesian situation.

Reviewer #4 comments:

1. Summary of article

The work is very interesting, novel and is well suited for publication in PLOS-ONE.

I have reviewed the author’s response to the previous reviewers’ comments and am happy to see that all relevant points were taken on board to improve the manuscript.

Au reply: We thank the reviewer for these positive remarks.

2. Introduction

Line – “EBV was first identified in cells of Burkitt Lymphoma (BL), an endemic cancer among sub-Saharan children, that is triggered by co-infection with Malaria parasites [reviewed in 8, 9]. BL is the most common cancer in children living in Malaria endemic regions in sub-Sahara Africa and Papua New Guinea [9,10],” - recent publications have been published highlighting new perspectives such as Ellis et al 2021 (https://doi.org/10.1186/s13027-021-00370-7), which can be referenced.

Au reply: The suggested extra reference’by Ellis et al., from 2021 contains an overview of data from 1753 publications during the last 20 years (2001-2020) on clinical-epidemiological and pathogenic-therapeutic aspects of Malaria and Cancer including BL. 

In our manuscript we named 6 references from recent 10 years on the pathogenesis and clinical aspects of BL in Malaria endemic regions (ref’s 9, 12, 23, 28, 29, 52) and 1 from 2005 (ref. 10), and we mentioned 2 recent reviews on Malaria and Cancer risk from 2017 (ref.13) and 2020 (ref. 31), which we feel cover all relevant aspects. Therefore, we prefer not to include any further references. 

3. Line – “There appeared to be a direct correlation between increases in plasma EBV viral load and progression of endemic Burkitt Lymphoma” abbreviate to BL.

Au reply: We checked and corrected all subsequent BL abbreviations.

4. In the introduction information on the four cytokines chosen to quantify in the study should be provided in greater detail beyond the summary paragraph provided. Further details on the selection/choice of specific cytokines used in this study is important. For example “in this study the amount of x,x,x, and x plasma cytokines was evaluated. X has a role in xx etc repeat.

Au reply: In the first round of reviewer’s comments and suggested revisions it was suggested to shorten the introduction regarding the role(s) of cytokines in acute Malaria, which we did. The suggested details and arguments for selecting the 4 cytokines and their proposed role in Malaria is now given in the discussion.

5. Materials and Methods/ Results and Discussion

Sample collection and Malaria parasite analysis 

– because of the age association with relationship between malaria and EBV it would have been useful for the field to have full malaria patient data i.e. positive parasites of P. falciparum (n=26), P. vivax (n=28), and mixed (P. falciparum and P. vivax) (n=14), or healthy controls (n=27) data presented in a table with age information to know for example how many patients with P. falciparum infection were under 5 etc. This should be reflected in the results and discussion. Whilst the gender could be of interest especially in any of the samples were from pregnant women, the age is of greater interest and should be included.

Au reply: These detailed patient data were also requested by reviewer #1 and are provided in the revised manuscript as supplemental information. This also includes the correlation analysis of patient’s age and gender against EBV and cytokine levels for each Plasmodium type infection and for the controls (Suppl. Figures 2 - 4). 

6. The authors should highlight any sample data from pregnant women.

Au reply: We did not use any sample from pregnant women.

7. Use of Kenya and Kenia in the manuscript change all “Kenia” to Kenya for continuity.

Au reply: This was checked and corrected throughout the manuscript.

---

## [Decision Letter · Decision Letter 2]

14 Dec 2021

Quantitative cytokine level of TNF-α, IFN-γ, IL-10, TGF-β and circulating Epstein-Barr Virus DNA load in individuals with acute Malaria due to P. falciparum, P. vivax or double infection in a Malaria endemic region in Indonesia.

PONE-D-21-19672R2

Dear Dr. Middeldorp,

We’re pleased to inform you that your manuscript has been judged scientifically suitable for publication and will be formally accepted for publication once it meets all outstanding technical requirements.

Kind regards,

Luzia Helena Carvalho, Ph.D.

Academic Editor

PLOS ONE

Additional Editor Comments (optional):

Reviewers' comments:

Reviewer's Responses to Questions

**Comments to the Author**

1. If the authors have adequately addressed your comments raised in a previous round of review and you feel that this manuscript is now acceptable for publication, you may indicate that here to bypass the “Comments to the Author” section, enter your conflict of interest statement in the “Confidential to Editor” section, and submit your "Accept" recommendation.

Reviewer #1: All comments have been addressed

Reviewer #3: All comments have been addressed

2. Is the manuscript technically sound, and do the data support the conclusions?

Reviewer #1: (No Response)

Reviewer #3: Yes

3. Has the statistical analysis been performed appropriately and rigorously? 

Reviewer #1: (No Response)

Reviewer #3: Yes

4. Have the authors made all data underlying the findings in their manuscript fully available?

Reviewer #1: (No Response)

Reviewer #3: Yes

5. Is the manuscript presented in an intelligible fashion and written in standard English?

Reviewer #1: (No Response)

Reviewer #3: Yes

6. Review Comments to the Author

Reviewer #1: (No Response)

Reviewer #3: The authors have addressed all the comments i made and the manuscript is very much improved. I have no additional comments.

7. PLOS authors have the option to publish the peer review history of their article (what does this mean?). If published, this will include your full peer review and any attached files.

Reviewer #1: No

Reviewer #3: No

---

## [Editor Report · Acceptance letter]

16 Dec 2021

PONE-D-21-19672R2 

Quantitative cytokine level of TNF-α, IFN-γ, IL-10, TGF-β and circulating Epstein-Barr Virus DNA load in individuals with acute Malaria due to *P. falciparum* or *P. vivax* or double infection in a Malaria endemic region in Indonesia. 

Dear Dr. Middeldorp:

I'm pleased to inform you that your manuscript has been deemed suitable for publication in PLOS ONE. Congratulations! Your manuscript is now with our production department. 

Kind regards, 

on behalf of

Dr. Luzia Helena Carvalho 

Academic Editor

PLOS ONE